# A Study of Purification in Pine Forest Soils after Salt Damage from the Tsunami in Enjugahama Beach, Wakayama Prefecture

**Hayato Masuda [1,\*] and Kyohei Yokota [2,\*]**

1 Advance Course of Ecosystem Engineering, National Institute of Technology, Wakayama College, Gobo 644-0023, Japan
2 Department of Civil Engineering, National Institute of Technology, Wakayama College, Gobo 644-0023, Japan
\* Correspondence: 2021e12@wakayama.kosen-ac.jp (H.M.); yokota@wakayama.kosen-ac.jp (K.Y.)

**Abstract:** The purpose of this study was to examine the feasibility of using a river as a water source for purification of soil damaged by the tsunami in a pine forest in Enjugahama, Mihama town, Wakayama prefecture, and to create a "purification map" visualizing the amount of purification water. Soil from the pine forest was placed in a plastic container and seawater was poured into it, followed by river water. The amount of water required for soil purification was determined by measuring the EC at this time. It was confirmed that 333,364 m$^3$ of water would be required to purify the entire pine forest, which is approximately 1 km$^2$. The time required to collect this volume of water from the West River would be 265 h (about 2 weeks) for an average flow rate. These results were aggregated to create the purification map. Using this map, it is possible to estimate the amount of water needed for purification at any given point and to make decisions, such as prioritizing areas that are easier to purify, thereby contributing to the purification of pine forests after tsunamis. However, it could be said that purification would be difficult in cases where seabed sediments have been deposited on the soil.

**Keywords:** pine forest; salt damage; purification; tsunami; humus

## 1. Introduction

On 11 March 2011, Japan was hit by the Great East Japan Earthquake, which had a magnitude of 9.0 and an intensity of 7 on the Japan Meteorological Agency seismic intensity scale. Tsunamis exceeding 7 m were observed in many areas. The maximum run-up height was observed to be 40.5 m [1]. On 15 January 2022, the eruption of Hunga volcano caused a tsunami that generated further tsunamis in many areas. Tsunamis occur in contexts other than earthquakes as well [2]. In the future, the Nankai Trough Mega Earthquake will definitely occur in Japan. It is assumed that it will have a magnitude of 9.1 on the Richter scale and a seismic intensity of 7. The seismic intensity in Wakayama prefecture is estimated to be 6 or 7 in the event of that massive earthquake. The tsunami height in the study area is estimated to be 18 m. Therefore, this height may greatly exceed the height of the tsunami resulting from the Great East Japan Earthquake [3].

The 2004 tsunami in the Indian Ocean and the 2011 tsunami generated by seismic motion from the Great East Japan Earthquake caused significant damage to pine forests and other vegetation growing in coastal areas. Many plants were affected to the extent that they were unable to regenerate [4]. Damage characteristics included branch breakage, trunk breakage, root turnover, overthrow, runoff, and standing mortality [5]. Pine trees are salt-tolerant plants. However, pine forests were severely damaged by the tsunami and have been difficult to restore. Efforts are underway in many areas to restore these plants and soils impacted by the tsunami [6]. Pine forests are effective as windbreaks and sand-control forests. They provide many benefits to people [7].

The possible effects of a tsunami include physical and chemical effects. Physical effects include being swept away by the force of the tsunami, such as in the case of an outflow.

This can cause extensive damage not only to vegetation in coastal areas but also to houses and other structures [5]. As for chemical effects, the typical examples are those caused by salt damage, such as standing water dieback [8]. Soil salinity immediately after the tsunami was well above the maximum value for crop growth (4 dS/m). It was reported to have a greater impact on the surface layer (1–2 cm deep) [9]. Thus, chemical impacts affect not only pine forests but also other plants. Basically, pine forests are salt-tolerant plants. However, pine wilt can occur when salinity exceeds a certain level [10]. In Hachinohe city, Aomori prefecture, Japan, there was no change in the pine forests due to the tsunami caused by the Great East Japan Earthquake for two months. However, red dieback occurred five months later. The pine trees may continue to grow without pine dieback if the concentration of soil salinity is reduced in some way within a certain period of time. Pine trees are basically tolerant of salt damage. However, red wilting occurs when the electrical conductivity is greater than 20 mS/m and the salinity is close to that of seawater (3.4%). The water elution method is often used as a method to remove salt from soils with high salt concentrations due to the effects of tsunamis and storm surges. Rainfall and irrigation can wash surface salts away from drainage channels. Thus, stripping saline topsoil is considered an effective salt removal measure that farmers can implement [9,11,12]. The principle is to remove salt from the soil through repeated dilution and permeation of the salt [13]. For the optimal method for salt removal, it has been reported that (1) a shorter water supply time and a longer suction time effectively reduce the amount of leachate and (2) too long a suction time increases the amount of leachate required. Thus, attempts have been being made to remove soil salts through rainfall. The salinity in the soil affected by the 2004 Indian Ocean tsunami was almost completely removed by rainfall during the rainy season. However, the salinity was reported to have had long-term effects on local crops. Therefore, it is preferable to use a certain amount of tap water or river water in addition to rainfall to remove salinity at an early stage.

One of the most important factors to keep in mind when removing salt from soil after a tsunami is the accumulation of bottom sediment [14]. The sediments from the oceanic bottom rolled up by the tsunami are deposited on the soil of the land. This bottom sediment contains many types of heavy metals and often also contains sulfur components. There are concerns about the emission of sulfuric acid and hydrogen sulfide from the bottom sediment. These are difficult to remove with rainfall. Therefore, there is a need to consider how to deal with them.

In this study, the remediation method for bottom sediment soils was investigated first of all. In addition, a method of removing salinity using river water was investigated. The amount of river water and the time required for salinity purification were investigated, including an examination of ways to secure that amount of water. The purpose of this project was to create a "purification map" that summarizes the results in a diagram. It will be possible to determine where to begin purification in the event of an actual disaster after creating this "purification map".

## 2. Survey Overview

### 2.1. About the Survey Site

Figure 1 shows a pine forest growing near Enjugahama Beach in Mihama town, Wakayama prefecture. This pine forest is located at the mouth of the Hidaka River, which flows into the central part of the Kii Peninsula. It is called a Japanese black pine grove and it is the largest in the Kinki region, with a maximum width of 500 m and a total length of approximately 4.5 km [14]. This pine forest was pslanted by Yorinobu Tokugawa, the first head of the Kii Domain, for tide protection.

The circled areas in Figure 1 were the soil-sampling points, and the entire pine forest was surveyed. This pine forest surrounds residential areas, schools, and hospitals. Therefore, the residents and the pine forest coexist in this area. The elevation of the pine forest in Enjugahama Beach is approximately 6 m, and the elevation of the residential area is approximately 10 m [15]. Part of this pine forest was planted on what was originally a

beach. Therefore, there are areas where the soil contains a large amount of gravel. In those areas, pine leaves have accumulated and the soil has turned to humus. Since the past shoreline is not known, it is unclear to what extent these areas correspond.

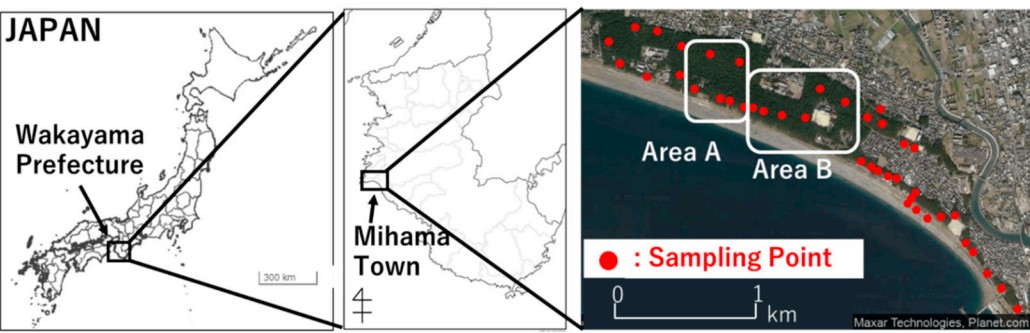

**Figure 1.** Survey sites.

### 2.2. Inundation Depth

Figure 2 shows the tsunami inundation depths in the event of the Nankai Trough Mega Earthquake. It is assumed that the Nankai Trough Mega Earthquake will have a magnitude of 9.1 on the Richter scale and a seismic intensity of 7. The seismic intensity in Wakayama prefecture is estimated to be 6 or 7 [3]. Figure 2 shows the inundation depths for Gobo city, Mihama town, and surrounding areas in Wakayama prefecture. The inundation depths exceed 10 m at some high points, indicating that the area is expected to be severely damaged. The tsunami inundation depths around Enjugahama Beach are estimated at 3–10 m, similar to those of the Great East Japan Earthquake.

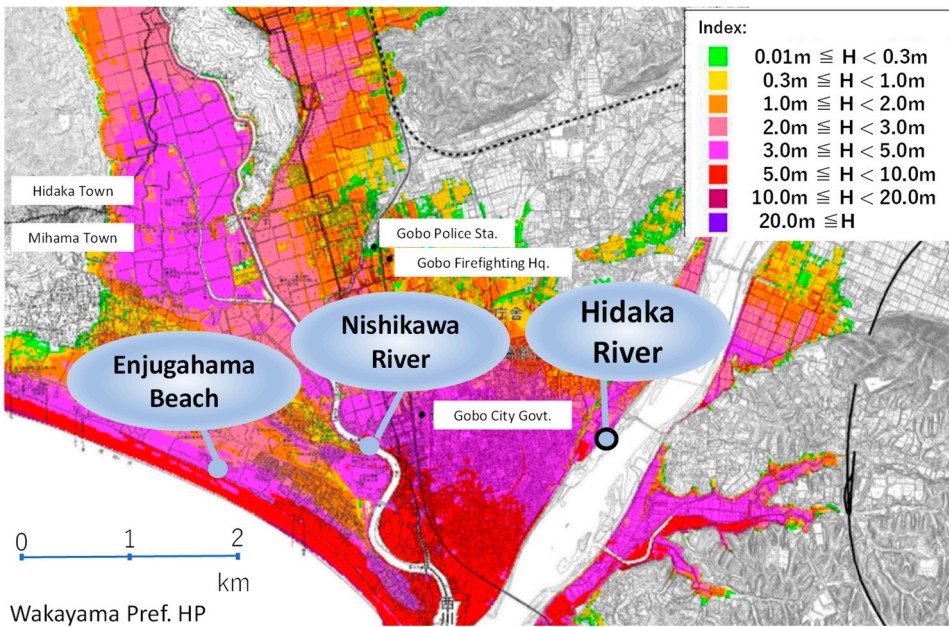

**Figure 2.** Inundation depths due to the expected tsunami (the Nankai Trough Mega Earthquake) and river water sampling points, this figure was used in accordance with the Terms of Use by Ref. [16]. Copyright 2013, wakayama prefecture. The original figure is Japanese. Therefore this figure was add the point names in English.

## 3. Survey and Experimental Methods

### 3.1. Soil Sampling

Soil samples were collected from the pine forest by digging up the soil with shovels. The soil was brought back to the laboratory for experiments. The period of the study was from June 2020 to November 2022.

### 3.2. Flow Rate

Figure 2 shows the locations where the flow rate was measured at the water sampling point. The cross-sectional area and flow velocity were measured and multiplied to calculate the flow rate. The cross-sectional area was measured using a total station (TI Asahi Co., Ltd., Saitama City, Japan, model number: PENTAX R-430N) and the flow velocity was measured using a flow meter (Tamaya Technics Inc., Tokyo, Japan, model number: UC-300V). A stage–discharge rating curve (HQ equation; H: water level measured in Wakayama prefecture, Q: water volume measured in the laboratory) was created to relate the measured flow rate to the water level data measured in Wakayama prefecture at the same time. Based on the formula and the water level data measured in Wakayama prefecture, the flow rate during the required period was calculated.

### 3.3. Soil Properties Tests

Particle size and water content tests were conducted to investigate soil properties. The particle size test was conducted according to the JIS standard (JIS A 1205, Japan standard) [17]. Stainless steel sieves (Sampo Corp., Koshigaya City, Japan, SUS304(18Cr-8Ni)) from Sampo Corporation were used. The sieve frame dimensions were 200 $\Phi \times 60$, and the sieve apertures were 4.75 mm, 3.35 mm, 2 mm, 0.85 mm, 0.425 mm, 0.25 mm, 0.106 mm, and 0.075 mm. The masses of the particles that remained on each sieve were measured with a mass meter to obtain the particle size. The moisture content ratio of the soil at each sampling point was tested according to the JIS standard (JIS A 1203, Japan standard) [18]. A constant-temperature dryer (Azwan, Ltd., Osaka City, Japan, model number: ON-300S) was used for drying, and the soil was dried at 110 °C for at least 24 h. The moisture content ratio was determined from the mass before and after drying.

### 3.4. Purification Experiment

As shown in Figure 3, a plastic container (JEJ Astage Corp., Sanjyou City, Japan, NF box) with internal dimensions W 153 $\times$ D 278 $\times$ H 165 mm (the width and depth are the size of the bottom) and a hole in the bottom was used. Several drainage sheets were laid down and soil collected at each site was placed in the box until it reached a height of approximately 150 mm. Then, 10 L of seawater was allowed to percolate into the soil. After all the seawater had been discharged, 1 L of river water was allowed to percolate in. The wastewater was collected in a separate container after 1 L of the river water had finished passing through the soil to be purified. The wastewater was measured by using a portable electrical conductivity measurement device (HORIBA, Ltd., Kyoto, Japan, model number: D-24). The unit of measurement used was mS/m. The infiltration of river water was allowed to continue until the EC stabilized while confirming the progress of purification. The method of operation can be summarized as follows. First, 150 mm of soil was placed in the container. Then, 10 L of seawater was poured into the soil, followed by 1 L of river water. The process was continued until the EC stabilized.

Next, purification experiments were conducted using containers of different sizes to determine the relationship between the size of the soil base area and the amount of water required for purification. The experimental procedure was the same as in the purification experiment described above. Containers with internal dimensions of 278 $\times$ 153 $\times$ 165 (7 L), 358 $\times$ 238 $\times$ 240 (24 L), and 439 $\times$ 309 $\times$ 300 (43 L) were used. The larger containers required more soil for the experiment, but it was not possible to secure large amounts of soil from the pine forest. As an alternative, silica sand sold by Tohoku Silica Sand Co was used. The names of the products used were No. 5 and No. 6. The soil was placed in the containers at depths of up to 150 mm and the amount of water required to complete purification was measured. The purification experiments were conducted in the same manner as above.

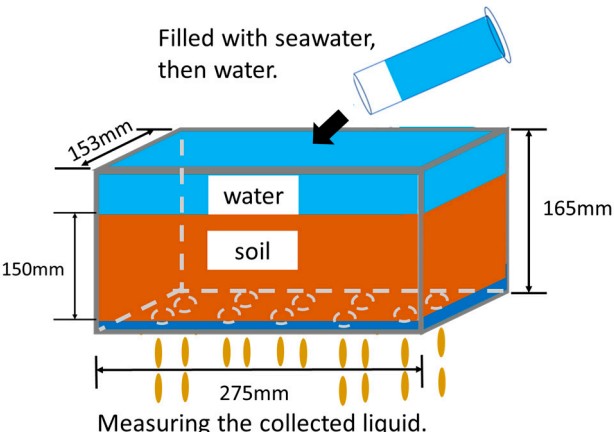

**Figure 3.** Method for the purification experiments.

*3.5. Analysis of Soil Composition*

The constituents of the water discharged from the soil during the purification experiment were analyzed using ion chromatography for $Na^+$ and $Cl^-$ and spectrophotometry for boron. $Na^+$ and $Cl^-$ were selected because the main component of seawater is NaCl and, thus, the influence of seawater could be more accurately determined [19]. Boron was selected because it is abundant in bottom sediments and environmental standards have been established. There are many studies that mention the presence of iron in bottom sediments as a problem. However, since iron is not included in the environmental standards, boron was selected as the priority in this study.

For the ion chromatography (DKK-TOA Corp., Tokyo, Japan, model number: ICA-2000) method, a mixture of methanesulfonic acid, sodium bicarbonate, and sodium carbonate was used as the eluent. For anion analysis, a chemical suppressor was attached to the chromatograph to remove cations, and sulfuric acid was used as the suppressor solution. The cations were analyzed with the non-suppressor method. Boron was analyzed with the spectrometric method using azomethine-H. The samples were placed in an absorption cell, and absorbance was measured at a wavelength of 410 nm using an absorption spectrophotometer (DKK-TOA Corp., Tokyo, Japan, model number: HACH DR6000).

## 4. Results

*4.1. Analysis of Soil Composition*

The pine forests at Enjugahama Beach include areas that were originally sandy beaches. Therefore, the nature of the soil in the pine forests varies greatly from place to place. Some areas have large pieces of gravel mixed in with the soil, some have humus and some have soil composed of fine-grained sand. Using soil from the pine forest, we measured the soil properties using the particle size test and water content using the water content ratio test. Figure 4 shows the measurement results. Soil properties were classified according to JIS standards. In the western part of the pine forest, sand and gravel (SG) were abundant, while sand (S) was mainly observed in the eastern part. Gravel sand (GS) was found in the central part. Moisture content tended to be higher in the western and central parts of the pine forest where the soil had decomposed, although this cannot be generalized since the dates of sampling were not the same.

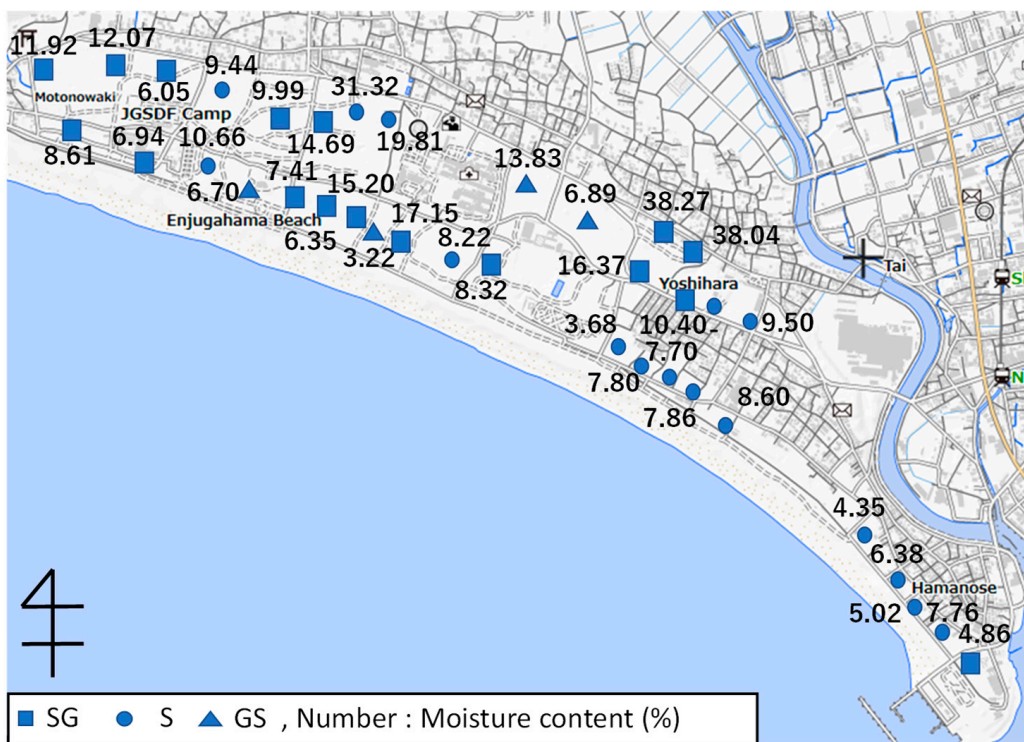

**Figure 4.** Distribution of soil properties and water content in the pine forest at Enjugahama Beach.

*4.2. Relationship between EC and Dissolved Constituents*

When large amounts of Na⁺ enter soils, plant nutrients, such as $Ca^{2+}$ and $K^+$, are replaced by $Na^+$ and healthy growth is inhibited. Therefore, salt removal is necessary to grow plants in soil after a tsunami. To verify the effectiveness of salt removal through remediation, the dissolved components ($Na^+$, $Cl^-$) were analyzed using ion chromatography in area B shown in Figure 1. The seawater contained potassium, magnesium, calcium, and sulfate ions. $Na^+$ and $Cl^-$ were particularly abundant compared with other components. Therefore, only $Na^+$ and $Cl^-$ are represented in Figure 1. The method used for the purification experiment is described below. Soil was placed in a perforated container (internal dimensions: W 153 × D 278 × H 165 mm) up to a height of 150 mm. After 10 L of seawater had percolated into it, river water was allowed to percolate through, and the dissolved constituents ($Na^+$, $Cl^-$) in the discharge water from the bottom of the container were measured (Figure 3). Changes in the concentrations of $Na^+$ and $Cl^-$ during the purification experiment are shown in Figure 5. These results were obtained per volume of soil used in the experiments. The soil volume was unified at 0.00638 m³. These results show that the concentrations of $Na^+$ and $Cl^-$ decreased with continued infiltration of water into the soil at the six analyzed sites. Figure 6 shows the relationship between EC and $Na^+$ and $Cl^-$. For $Na^+$ and EC, the approximate curve was y = 4.14x − 193.1. The correlation coefficient was 0.969, indicating a strong correlation between $Na^+$ and EC, while the correlation coefficient between $Cl^-$ and EC was 0.987, with an approximate curve of y = 2.57x − 130.1. The correlation coefficient between $Cl^-$ and EC was 0.987, which indicates a strong correlation. These results suggest that EC trends in remediation experiments can be used as an indicator of the progress of salt removal from seawater-affected soils.

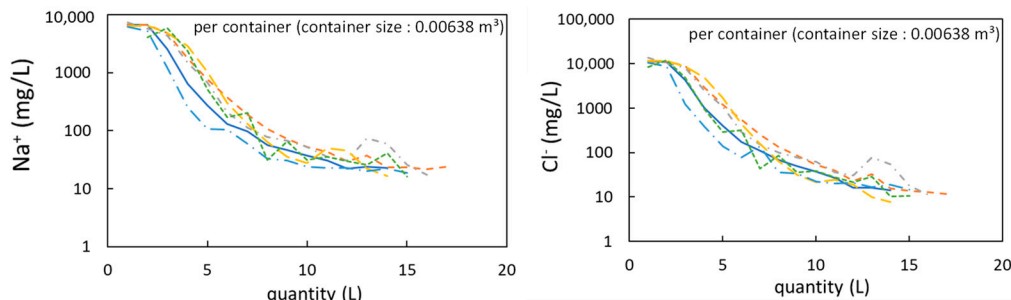

**Figure 5.** Analysis results for $Na^+$ and $Cl^-$ using ion chromatography at six sites in area B.

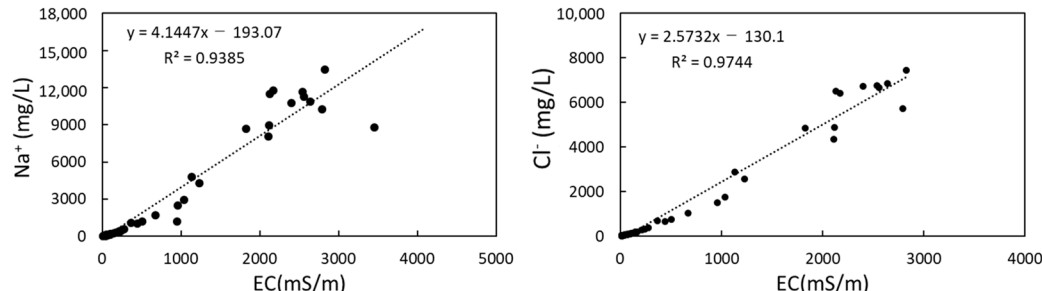

**Figure 6.** Relationship between dissolved constituents and EC.

### 4.3. Impact of Sea Sediment

The tsunami is expected to deposit bottom sediment in the soil to be purified [14]. Therefore, we verified whether it is possible to complete purification even when bottom sediment is deposited. Figure 7 shows the EC, time, and boron from the purification experiments. These results were obtained per volume of soil used in the experiments. The soil volume was unified at 0.00638 $m^3$. The amount of water required for purification of the bottom sediment was about 75 L. This is a large amount of water compared to the purification of soil in pine forests. It was also found that infiltration required more time. These factors may have been due to the small particle size of the bottom sediment, which made it difficult for water to percolate through the soil. The concentration of boron decreased with decreasing EC, but 45 L was not enough to reduce the concentration to 1 mg/L, which is the environmental standard value. This value of 45 L was larger than the amount of water required for purification of soils in pine forests, which will be discussed later. Based on the above, it is considered necessary to remove the bottom sediment through excavation or other means, as purification with river water or tap water is not suitable.

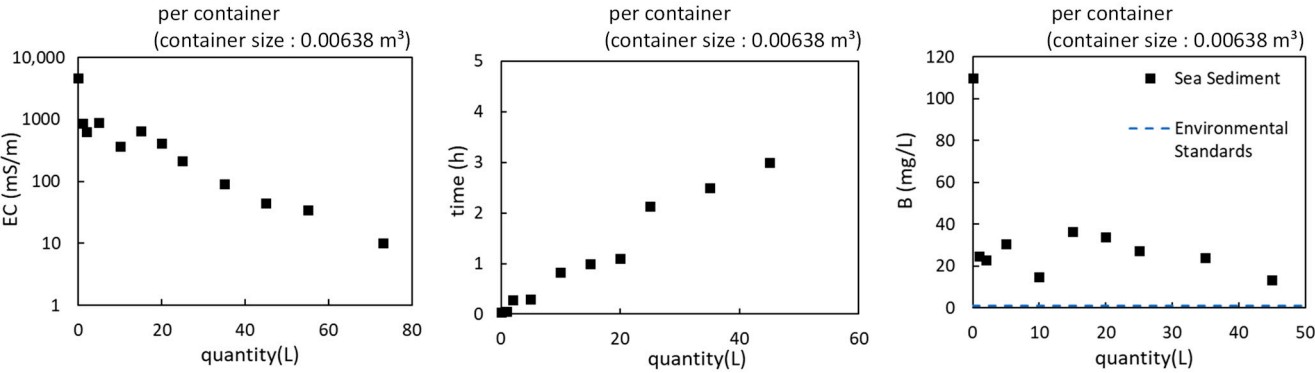

**Figure 7.** Results of bottom sediment purification and absorbance analysis.

### 4.4. Relationship between the Area to Be Purified and the Amount of Water Required for Purification

In all purification experiments, an NF box (W 278 mm × D 153 mm × H 165 mm) was used. When considering the actual purification of land, it is necessary to extend the experimental results to estimate the amount of water required for soil purification. However, the relationship between the results obtained from experiments using smaller containers and their extension to the soil of a pine forest was unclear. Therefore, we investigated how the results would change when the containers were enlarged. Although a large amount of soil was needed for this experiment, it was decided to avoid collecting soils from the pine forest for conservation reasons. Instead, Tohoku Silica Sand No. 5, which has a similar grain size distribution as that of the soil in the pine forest, was used. We also examined the relationship for Tohoku Silica Sand No. 6, which has a different grain size distribution. Table 1 shows the average grain size distribution for Enjugahama Beach soil and the grain size distributions for Tohoku Silica Sand Nos 4 to 7 and standard sand (Toyoura Silica Sand). The average Enjugahama Beach pine forest soil had a grain size of 0.425 mm or larger, which accounted for 90.7% of the soil. Tohoku Silica Sand No. 4 and No. 5 had similar grain size distributions as the pine forest soils. After comparing Tohoku Silica Sand No. 4 with the pine forest soils, it was not adopted because it contained a similar proportion of grains 0.425 mm in size but no grains smaller than 0.3 mm. Tohoku Silica Sand No. 5 was used in the experiment because it contained the smallest grain sizes (up to 0.106 mm) and was judged to be more similar to the pine forest soil than No. 4. Figure 8 shows the relationship between the size of the bottom area of the container and the amount of water required for purification. The approximate formula for Tohoku Silica Sand No. 5 was y = 0.0317x, with a strong correlation coefficient of 0.972. The approximate formula for Tohoku Silica Sand No. 6 was y = 0.0352x, with a correlation coefficient of 0.997. Since similar trends were confirmed for Tohoku Silica Sand No. 5 and No. 6, this relationship was considered to be unaffected by the grain size distribution. Therefore, it was considered that the amount of water required for purification could be obtained by extending the area of the pine forest in a similar manner.

**Table 1.** Particle size distributions of soils in the pine forest and silica sand.

| Name | Particle Size (mm) | | | |
| | 0.425 | 0.3(≈0.25) | 0.106 | 0.075 |
|---|---|---|---|---|
| Average pine forest soils | 90.7 | 7.5 | 0.9 | 1.0 |
| Tohoku Silica Sand No.4 | 99.6 | 0.4 | 0 | 0 |
| Tohoku Silica Sand No.5 | 79.5 | 16.1 | 4.4 | 0 |
| Tohoku Silica Sand No.6 | 17.4 | 50.1 | 32.3 | 0.2 |
| Tohoku Silica Sand No.7 | 0 | 0.6 | 92.8 | 6.6 |
| Standard Sand (Toyoura Silica Sand) | 8.92 | 90.37 | 0.45 | 0.24 |

(Leftmost row label: particle size distribution (%))

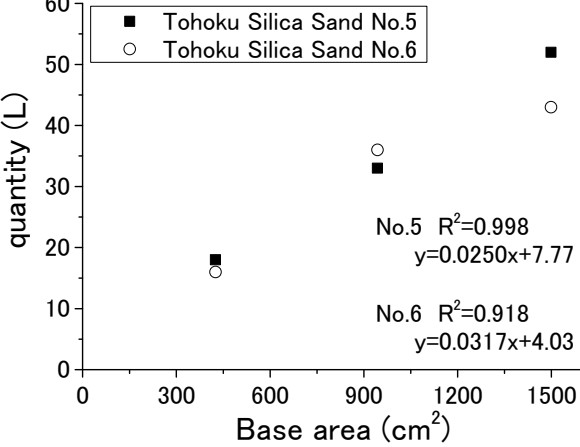

**Figure 8.** Relationship between the size of the bottom area of the container and the amount of water required for purification.

### 4.5. Results of Purification Experiments with Pine Forest Soils

#### 4.5.1. Amount of Water Required for Purification of the Pine Forest

Purification experiments were conducted at a total of 41 sites (Figure 1). Figure 9 shows the EC for the nine sites selected for the purification experiments. These were selected for their soil properties (sand and gravel (SG), sand (S), and gravel sand (GS)). If there had been a relationship between soil properties and EC, the amount of water required for purification could have been estimated from the soil properties, but multiple regression analysis did not show this relationship. The results of the multiple regression analysis are shown in Table 2.

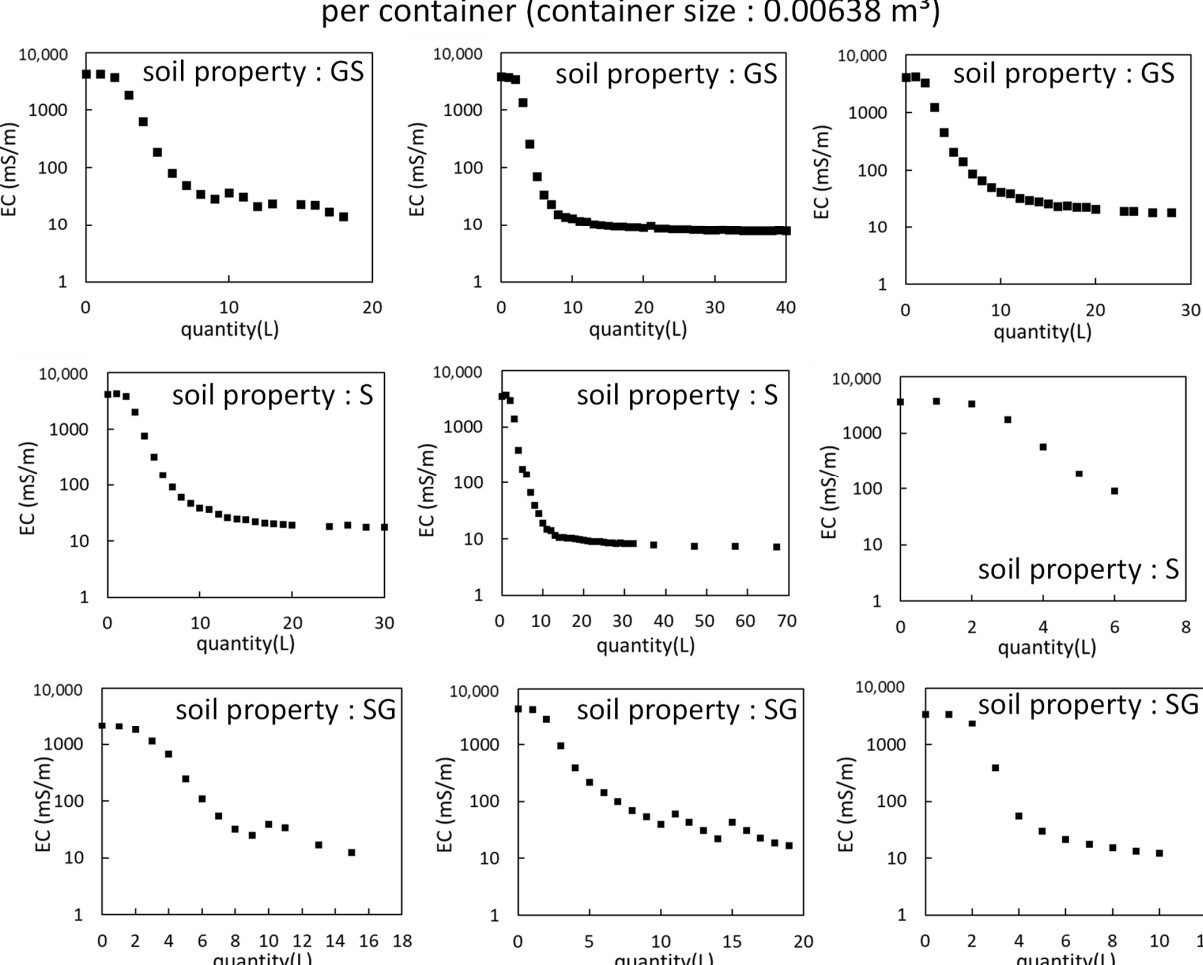

**Figure 9.** EC from purification experiments.

**Table 2.** Results of multiple regression analysis of soil properties and the amount of water required for purification. #NUM! means that the number is small.

|  | Facter | SEI | t | *p*-Value |
|---|---|---|---|---|
| intercept | 13.562 | 3.590 | 3.778 | 0.001 |
| gravel fraction (%) | 0 | 0 | 65535 | #NUM! |
| sand fraction (%) | 0.010 | 0.044 | 0.238 | #NUM! |
| fine-grain fraction (%) | 2.991 | 1.191 | 2.511 | 0.017 |

The purpose of this study was to estimate the amount of water required for purification of the soil across the entire Enjugahama Beach pine forest and to create a purification map. In order to do this, the soil in the vast pine forest had to be surveyed in a uniform manner.

However, there were some locations, mainly in area A in Figure 1, where there was a lack of data on EC reduction during the purification experiment. Therefore, we selected a function that could approximate the results of the purification experiment using the analysis software Origin/pro (Lightstone Corp., Tokyo, Japan, version 2022) to simply supplement the data. The following is a description of the data completion procedure. Function models were ranked based on the results from the sites where sufficient data were available, and functions suitable for approximation were determined. The Akaike information criterion (AIC) and Bayesian information criterion (BIC) were used as indices for comparing models. Smaller values for these indices indicated the suitability of the function used for approximation. The graphs of EC and water volume, as shown in Figure 9, were ranked. As a result, "Logistic", shown in Equation (1), was the most suitable function for approximation.

$$y = \frac{A_1 - A_2}{1 + \left(\frac{x}{x^0}\right)^p} + A_2 \tag{1}$$

where y is the EC (mS/m), x is the water quantity (L), and $A_1$, $A_2$, $x^0$, and $p$ are constants.

This function was used to complement the data by adapting it to locations with insufficient data. Figure 10 shows the change in the results due to the completion of the data. These results were obtained per volume of soil used in the experiments. The soil volume was unified at 0.00638 m$^3$. The function ex.2 was selected, and the slope of the graph at the end of purification was calculated as the dashed line overlapping ex.1. We determined completion of purification that the slope of ex.1 reached when the slope of ex.2 became that value. According to this, the soil of ex.1 could be considered to be purified with 15 L. Thus, the amount of water required for purification could be estimated simply by using Equation (1). The points corresponding to ex.1 were treated as the results of the purification experiments based on the approximation by Equation (1).

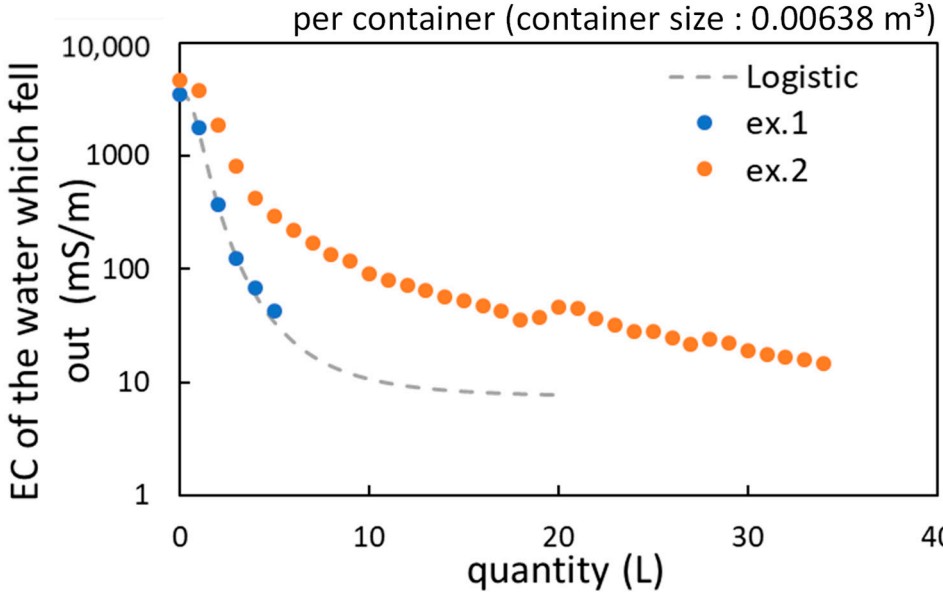

**Figure 10.** Comparison of EC transitions in purification experiments with and without data completion (ex.1: measured values at sites where data completion was required, ex.2: measured values at sites where data completion was not required).

Figure 11 was prepared using the quantity of water required for purification and the coordinates of the points where the samples were collected. Figure 11 shows the time required for purification per volume of soil used in the experiment (0.00638 m$^3$), with symbols between 6.7 L and 30 L at approximately 2 L intervals. The contour map shown in Figure 12 was prepared based on Figure 11. When creating Figure 12, it was necessary

to use a color scheme only for the areas where pine forests were located. Therefore, the outer frame was created by obtaining the coordinates of the outer shape of the pine forest. The contour map shows that purification in the western part of the pine forest, which is dominated by sand and gravel (SG), would require nearly 30 L of water at most, while purification in the central part, which is dominated by sand (S), especially in area A, would require less than 10 L. In the eastern part, the amount of water required for purification would be between 10 L and 28 L, indicating that purification could be completed with the amount of water between the western and central parts of the forest. This suggests that the soils in the western part of the pine forest are less suitable for purification than the other soils in terms of the amount of water required. Figure 13 shows the calculated amount of water required for purification of the pine forest soils for each of the classes in Figure 12. The horizontal axis is the amount of water required for purification, and the vertical axis is the amount of water required for the total area for each class in Figure 12. The total from these figures indicated that 333,364 m$^3$ of water would be required for the purification of the entire pine forest.

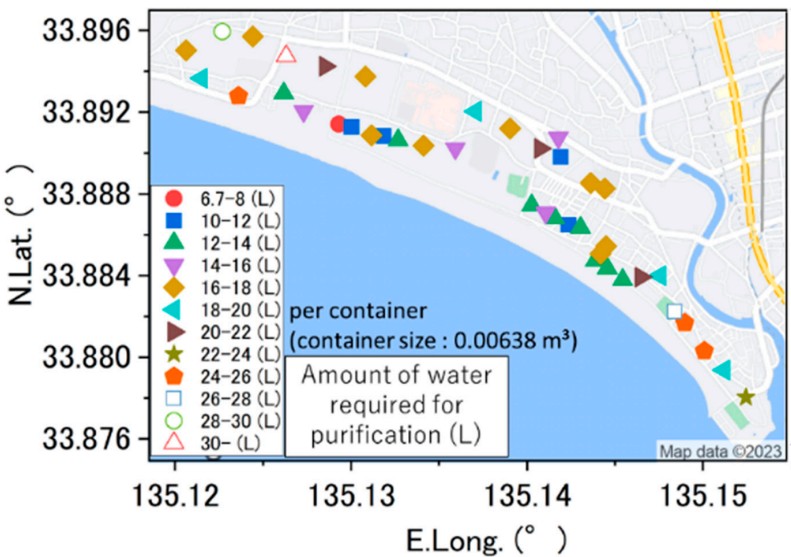

**Figure 11.** Distribution of amounts of water required for purification.

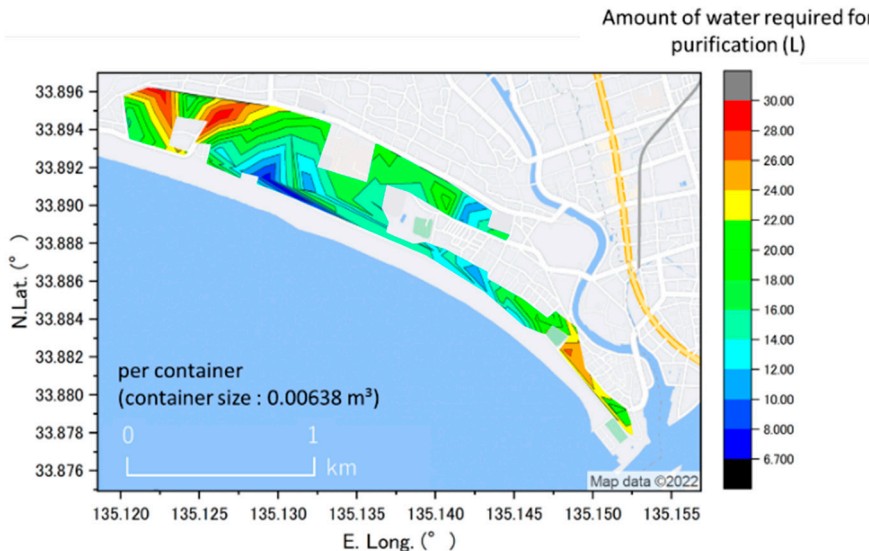

**Figure 12.** Contour map of the amounts of water required for purification.

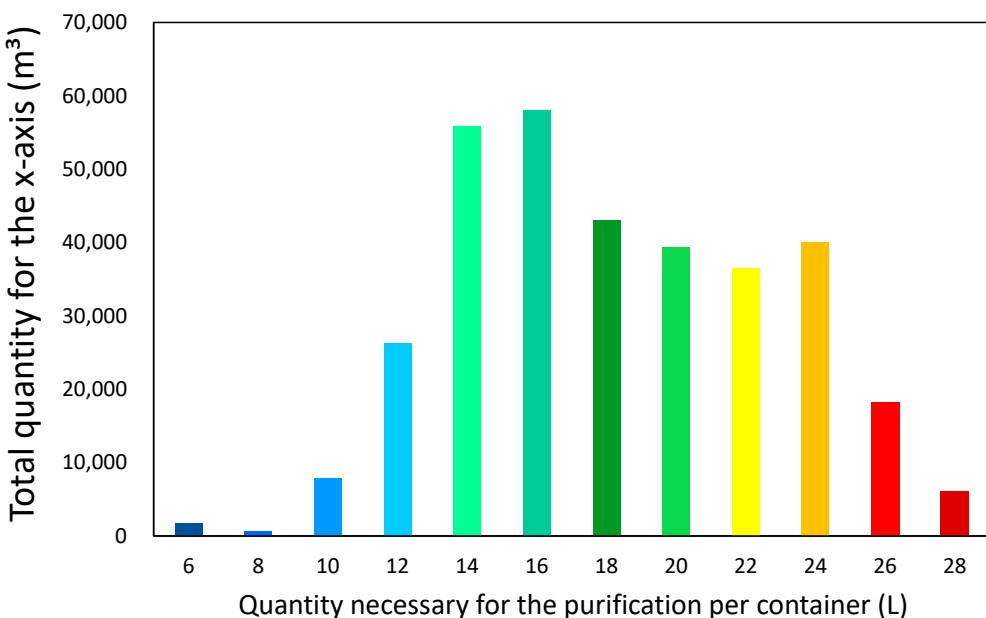

**Figure 13.** Total water volume calculated from the area of each class in the contour map.

4.5.2. Time Required for Purification of the Pine Forest

The permeability times obtained from the purification experiments are shown in Figure 14. These were selected for each soil type (sand and gravel (SG), sand (S), and gravel sand (GS)). As in the previous section, Origin2023 was used to rank the models for the graphs represented in Figure 14. As a result, "allometric1", shown in Equation (2), was the optimal model. Integration was performed for each of the water quantity intervals determined in the previous section. This was taken as the time required for purification. Figure 15 shows the results of the data completion. Ex.2 was used to supplement ex.1. The time required for 1 L of water to permeate was calculated and is shown as the dashed line.

$$y = ax^b \tag{2}$$

where y is time (h), x is water quantity (L), and a and b are constants.

Figure 16 was prepared using the time required for purification determined from the purification experiments and the data completion and the coordinates of the sampling points. Figure 16 shows the time required for purification, which is indicated on the map with symbols at 2.5 h intervals from 0.1 h to 32.5 h. As in the previous section, the contour map shown in Figure 17 was created based on Figure 16. Figure 17 shows that purification would be completed within 5 h in the eastern and central parts of the pine forest. In contrast, in the western part of the pine forest, purification would take more than 30 h in some areas. Area A had the same low value as in the previous section because the soil in this area has not been converted to humus and is close to the sandy beach, which was thought to facilitate purification due to its high permeability. On the other hand, it is also possible that the values calculated were lower than the results obtained from the experiment due to the complementation by the function.

**Figure 14.** Time in purification experiments.

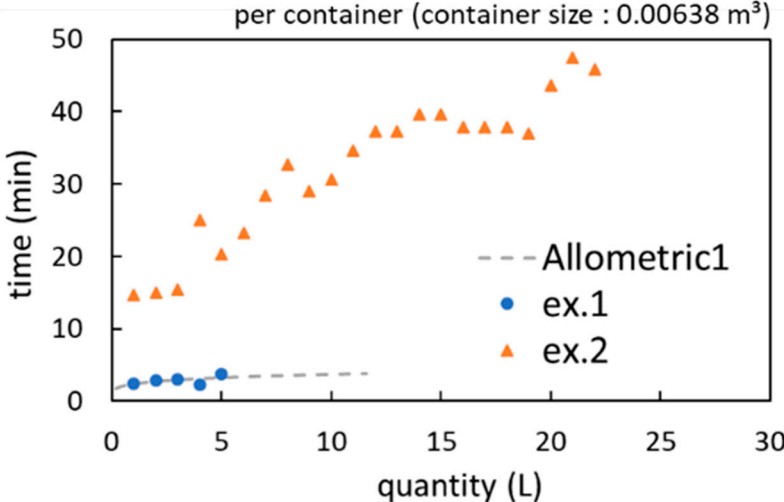

**Figure 15.** Comparison of time transitions in purification experiments with and without data completion (ex.1: measured values at sites where data completion was necessary, ex.2: measured values at sites where data completion was not necessary).

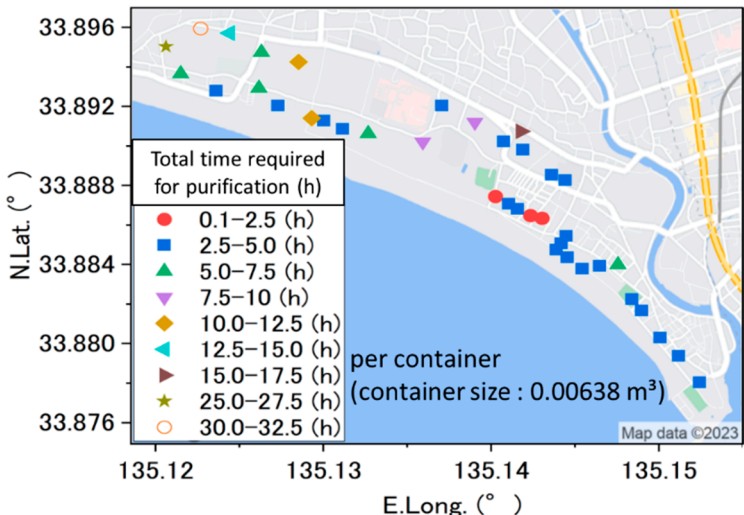

**Figure 16.** Time required for purification.

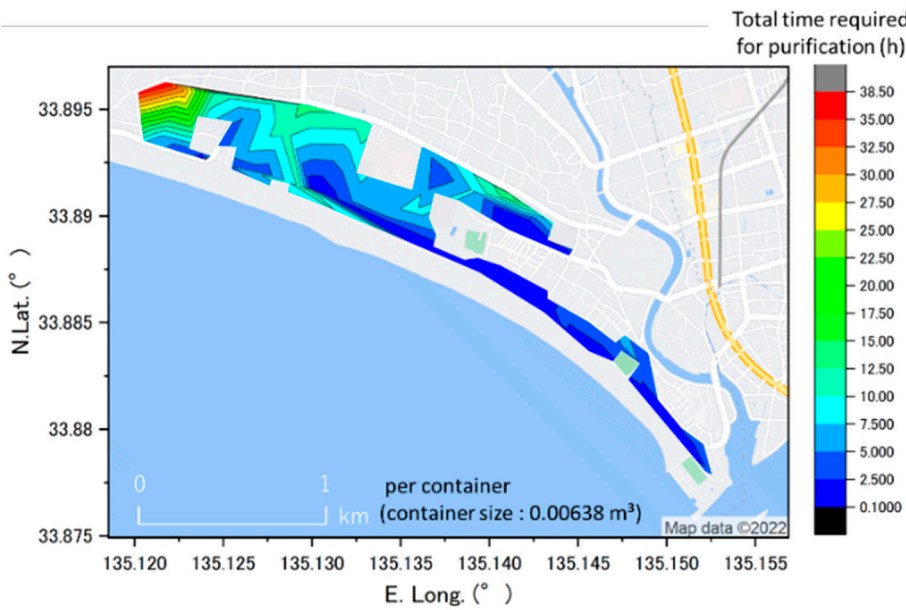

**Figure 17.** Contour map of time required for purification.

### 4.6. Flow Rate of The Nishigawa River

As mentioned above, the amount of water required to complete the purification of the pine forest at Enjugahama Beach was estimated to be 333,364 m$^3$. It is not easy to secure this amount of water. In addition, this would be a post-disaster mission. It is easy to imagine that water shortages will occur, and we cannot rely on the water supply as our water source in the midst of such shortages. For this reason, we assumed that water from the Nishikawa River, which flows through Mihama town, would be pumped for purification. However, as shown in Figure 2, the Nishikawa River is also expected to be damaged by the tsunami. Therefore, we considered taking water samples upstream of the Nishikawa River, where it is not expected to be inundated by a tsunami. Figure 18 shows the relationship between the flow rate we measured upstream of the Nishikawa River and the water level measured in Wakayama prefecture. The stage–discharge rating curve shown in Figure 18 is expressed by Equation (3).

$$y = (6.09 \pm 0.06)x^{3.69 \pm 0.06} \qquad (3)$$

where x is the water level (m) and y is the flow rate (m$^3$/s).

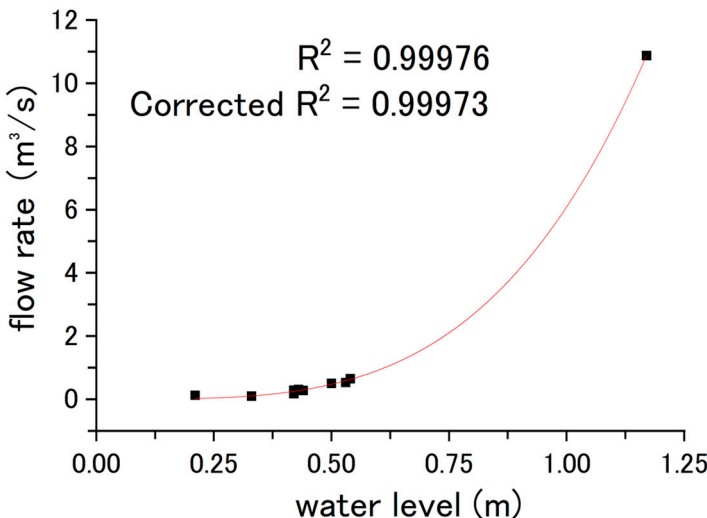

**Figure 18.** Stage–discharge rating curve upstream of the West River.

From the water level data measured in Wakayama prefecture, the annual mean water level of the Nishikawa River from April 2021 to May 2022 was 0.46 m and the minimum water level was 0.1 m. Using the stage–discharge rating curve, it was found that the average annual flow rate of the Nishikawa River is 0.35 m$^3$/s and the minimum flow rate is 0.0012 m$^3$/s. Assuming that the amount of water (333,364 m$^3$) required for purification of the entire pine forest (approximately 1 km$^2$) would be secured from the upper reaches of the Nishikawa River, the average time required would be 265 h (approximately 2 weeks), although the flow rate fluctuates depending on the season and other factors. On the other hand, considering the case of minimum flow, it was calculated that it would take 75,245 h (8.5 years) to recover the river water. Pine forests die within about three months, so the upper reaches of the Nishikawa River that were used as a sampling point may not be able to serve as a water source during periods of low precipitation.

Equation (4) shows the time required for purification to be completed after the tsunami. If purification is conducted under average river flow conditions, it can be expected to be completed within 3 months of the pine tree die-off if only the work of securing the river water and spreading it on the soil is considered. However, it is difficult to predict how many days after the disaster the forest will have recovered to the point where this plan can be implemented, and it is difficult to estimate the time required to remove the sediments. In addition, the specific means of collecting and transporting water have not been determined at this stage. Furthermore, it is necessary to consider other water sources as backups in case purification cannot be completed in time using only the Nishikawa River as the water source due to factors such as low precipitation, delays in the start of purification work, or the Nishikawa River becoming unusable for some reason. Various other factors will affect the purification plan for the pine forest. At present, it is not possible to devise a plan that takes all of these factors into account.

$$T = Tr + Tm + Tp + To \tag{4}$$

T: Time from the occurrence of the tsunami disaster to the completion of purification;
Tr: Time required to start the purification plan;
Tm: Time required for the removal of sediments, such as sea bottom mud;
Tp: Time required for purification work;
To: Time required for other factors.

### 4.7. Creation of the Purification Map

Based on the above, a purification map was created to facilitate the purification of soils in the pine forest after the tsunami (Figure 19). This purification map assumes that purification will take place when the bottom sediment in the sea area has been removed. The purification map shows the target area for purification, the year it was created, the source of the river water, the contour map from Figure 12 that was reorganized, what can be done with the purification map, and some notes. The depth of the soils to be purified was set at 0.15 m. The purification map can be used to estimate the amount of water required for purification in a given area. When considering the purification of large areas of land at low cost, priority can be given to areas where the amount of water required is low. When securing river water at the sampling points shown in the purification map, it is important to note that, during periods of low precipitation, it may not be possible to secure sufficient river water for purification. The purification map was created for the pine forest at Enjugahama Beach, but it should be possible to create purification maps for other coastal forests using the same method.

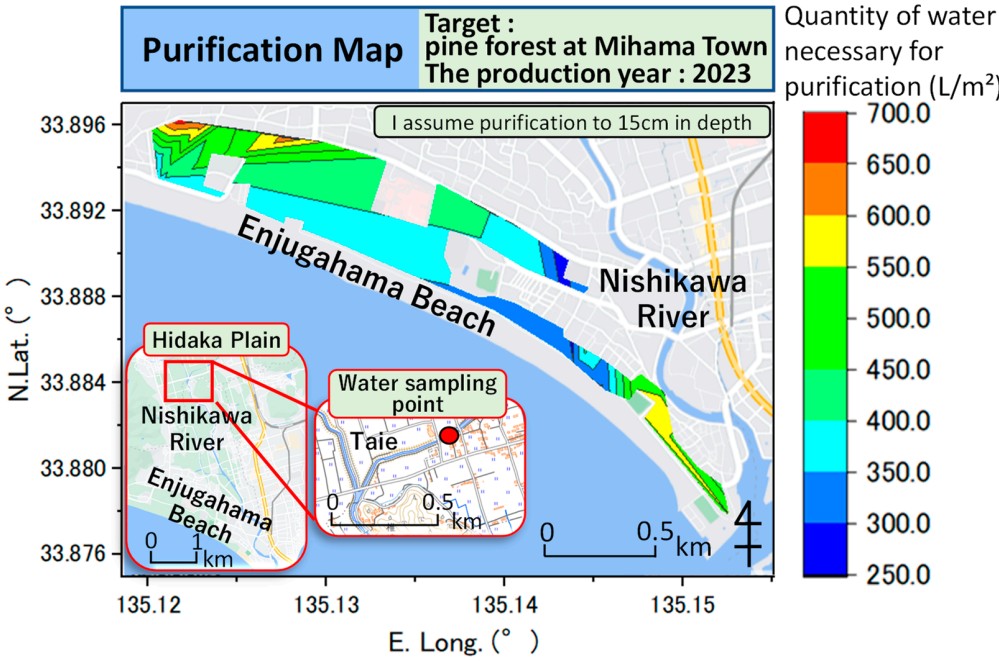

**Figure 19.** How to use the purification map.

## 5. Discussion

The amount of water required for soil purification was found to be proportional to the volume of soil. It was feasible to apply the system to actual sites using the results of

experiments conducted with smaller containers because of the actual use of the pine forest soil in this study. The results were used to create a map of the amount of water needed for purification of the pine forest soil, and the purification map showed the amount of water needed in different districts. It was assumed that it would be supplied from a river that flows nearby, which would provide a way to secure that amount of water. This would eliminate the need to use tap water, and the only cost would be for the power source to transport the river water. The pine forests are vegetated at around 10 m above sea level, and the maximum distance from the river to the pine forests is approximately 5 km. A power source capable of handling this would be needed. The flow rate of that river is 0.35 $m^3$/s on average and 0.0012 $m^3$/s at a minimum. According to our calculations, a total of 265 h (about 2 weeks) would be required on average, although the flow rate fluctuates depending on the season and other factors, in the case that the amount of water required to purify the entire pine forest were secured from the Nishikawa River. According to a report, there was a case of pine wilt five months after the Great East Japan Earthquake (11 March 2011). There would be enough time to prevent pine wilt if the purification of the pine forest were completed within this two-week period. It is important to note with regard to using river water for purification in this way that, when bottom sediment from the seafloor was deposited on the soil by the tsunami, the concentration of heavy metals in this bottom sediment could not be reduced to below the standard value, even when over 70 L of water was used. This value was very high compared to the amount of water needed to purify the soil in the pine forest. Based on these facts, bottom sediments are not suitable for purification by river water or tap water. The purification of the pine soil should be undertaken with river water after removing the bottom sediment when bottom sediment is deposited on soil by a tsunami.

## 6. Conclusions

The purpose of this study was to investigate the possibility of purification of the pine forest at Enjugahama Beach in Mihama town, Wakayama prefecture, after damage from a tsunami using the Nishikawa River as a water source and to create a purification map visualizing the amount of water needed for purification. The results of the experiment showed that purification is difficult when bottom sediment is deposited. Therefore, it is necessary to remove the sediments by some means if they have piled up. The amount of water needed to purify the entire pine forest was considered to be 333,364 $m^3$. It was assumed that it would take approximately 2 weeks to collect this amount of water from the Nishikawa River under average flow conditions and as long as 8.5 years under low flow conditions. Under average flow conditions, purification could be completed within 3 months of the onset of pine die-off. However, during periods of low flow, purification using only the upper reaches of the Nishikawa River as a water source would be difficult, and other sources, such as other rivers or ponds, should be considered. The time from the tsunami disaster to the completion of purification includes the time required to start purification, the time required to remove sea sediment, and the time required for the purification process. With these time limitations, more than adequate measures need to be taken in advance to prevent pine tree mortality. One such measure would be to improve the drainage system of the pine forest in order to accelerate the purification process. A purification map was developed to facilitate the purification process in an efficient and rational manner. The purification map included a contour map showing the amount of water required for purification and the water sampling points. The purification map enables the estimation of the amount of water required for purification at a given point and can help in making decisions to give priority to areas that can be easily purified. The purification map can also be used to remove salt from a site. For example, the purification map could be useful in the case of salt damage caused by a large typhoon splashing seawater into the target area.

In the Great East Japan Earthquake, many coastal forests were washed away by the tsunami. Most of the trees that escaped conversion to driftwood were destroyed by seawater

percolation. These areas are now being reborn as new towns. When a city is rebuilt after a disaster, such as a tsunami, there are likely to be more than a few people who want to restore the city to its original state. What if the pine forests that were a part of daily life were to disappear? It is too late to start thinking about soil purification after a disaster has occurred. It is very sad to see the loss of pine forests that have been a part of our daily lives. It is too late to consider purification after a disaster has occurred. It is necessary to take all possible precautions in advance. We hope that this study will serve as an option for urban development after disasters.

**Author Contributions:** Writing—original draft preparation, K.Y.; Other, H.M. All authors have read and agreed to the published version of the manuscript.

**Funding:** This research received no external funding.

**Informed Consent Statement:** Not applicable.

**Data Availability Statement:** All of the research data published in this report is owned by YOKOTA Lab in National Institute of Technology, Wakayama college.

**Conflicts of Interest:** The authors declare no conflict of interest.

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
