# Peer review of "A Study of Purification in Pine Forest Soils after Salt Damage from the Tsunami in Enjugahama Beach, Wakayama Prefecture"

_sustainability, doi:10.3390/su15119136_

Round 1
Reviewer 1 Report
Manuscript comprises brief informative introduction about tsunami impact on coastal areas, clear description of studied region including spatial distribution of inundation depth and soil texture, laboratory leaching experiments and their results, models for approximation of EC reduction during the purification experiment, calculations of water amount and time required for purification at different sites, so called Purification Map created on the bases above mentioned information.
The authors demonstrate approach for evaluation possibility of purification of pine forest at coastal area which was damaged by the tsunami using river water. They propose to create a Purification Map that visualizes amount of water required for purification of coastal soils salinized by sea waters after tsunami.
The manuscript clear describes all steps for creation of Purification Map.
Notes:
1. Row 198. To delete words “… and figure 6”.
2. Row 201. Right reference is “Figure 6 …” instead “Figure 7 …”.
3. Row 202. Right reference is “Figure 6 shows…” instead “Figure 8 shows…”.
4. Rows 215-229. Would the authors like to submit soil texture for deposited bottom sediments that was used for leaching experiment? Would the authors like to submit infiltration coefficients of soils in pine forest and deposited bottom sediments that was used for leaching experiments?
5. Figure 8. Point (0,0) at the beginning of figure 8 is not clear. It cannot be experimental. It will be better to exclude it from the figure and to write that linear regression was calculated without intercept.
6. Rows 307-308. Figure 10 is absent.

Reviewer 2 Report
The manuscript by Masuda and Yokota investigated the feasibility of using a river as a water source to purify soil damaged by the tsunami in a pine forest and created a purification map to estimate the amount of water needed for purification. The results showed that a large amount of water was required, and purification might be difficult when seabed sediments were deposited on the soil.
In summary, the article is logically consistent, and the methods used are appropriate. I agree it is of scientific significance. However, as an academic paper, it lacks a Discussion Section. It would be beneficial to include a section discussing the limitations of the study, such as potential differences between experimental and natural conditions. There are also some deficiencies in the introduction regarding the explanation of tsunami-related content. Hence, a minor revision should be conducted carefully as follows.
(1) In the Introduction, it would be helpful to briefly mention the tsunami risk in Wakayama Prefecture and recent tsunami events, such as the 2022 Tonga volcanic tsunami.
Reference: https://doi.org/10.1785/0220220098
(2) Line 97: There is a lack of information on the Nankai megathrust earthquake. Is this analysis based on a hypothetical scenario or the results of a Probabilistic Tsunami Hazard Assessment (PTHA)? If it is the former, please provide details on the earthquake source and magnitude. If it is the latter, please provide a brief overview of the calculation process.
(3) Figure 2: Please add a scale bar to the map.
(4) Line 139: Please provide more information on the electrical conductivity measurement process? How was it conducted and what were the measurement units used?
(5) Line 192: What other constituents, besides Na+, Cl-, and boron, were present in the discharged water from the soil during the purification experiment? Do you consider potassium ions in seawater?
(6) This paper lacks a Discussion Section. The heavy metals present in the bottom sediment are resistant to removal by rainfall. However, is it possible that rainfall could still have some impact on their concentration levels? It would be beneficial to further elaborate on this in the discussion section.
(7) The manuscript would benefit from a discussion of potential differences between laboratory experiments and fieldwork in the natural environment. It would also be helpful to address any limitations or biases that may have affected the experimental results.
(8) As a practical issue, it is necessary to discuss the cost of purification. Please include this aspect in the Discussion section.
